# 3D-Vision-Transformer Stacking Ensemble for Assessing Prostate Cancer Aggressiveness from T2w Images [note 1]

**DOI:** 10.3390/bioengineering10091015

**Published:** 2023-08-28

**Authors:** Eva Pachetti, Sara Colantonio

**Affiliations:** 1“Alessandro Faedo” Institute of Information Science and Technologies (ISTI), National Research Council of Italy (CNR), 56127 Pisa, Italy; sara.colantonio@isti.cnr.it; 2Department of Information Engineering (DII), University of Pisa, 56122 Pisa, Italy

**Keywords:** vision transformers, ensemble, prostate cancer, MRI imaging, deep learning, classification

## Abstract

Vision transformers represent the cutting-edge topic in computer vision and are usually employed on two-dimensional data following a transfer learning approach. In this work, we propose a trained-from-scratch stacking ensemble of 3D-vision transformers to assess prostate cancer aggressiveness from T2-weighted images to help radiologists diagnose this disease without performing a biopsy. We trained 18 3D-vision transformers on T2-weighted axial acquisitions and combined them into two- and three-model stacking ensembles. We defined two metrics for measuring model prediction confidence, and we trained all the ensemble combinations according to a five-fold cross-validation, evaluating their accuracy, confidence in predictions, and calibration. In addition, we optimized the 18 base ViTs and compared the best-performing base and ensemble models by re-training them on a 100-sample bootstrapped training set and evaluating each model on the hold-out test set. We compared the two distributions by calculating the median and the 95% confidence interval and performing a Wilcoxon signed-rank test. The best-performing 3D-vision-transformer stacking ensemble provided state-of-the-art results in terms of area under the receiving operating curve (0.89 [0.61–1]) and exceeded the area under the precision–recall curve of the base model of 22% (*p* < 0.001). However, it resulted to be less confident in classifying the positive class.

## 1. Introduction

In 2020, prostate cancer (PCa) was the world’s second-most-common tumor among men (accounting for 14.1% of new diagnoses, just behind lung cancer) and the fifth for mortality (6.8%) [1]. For screening PCa, physicians primarily use the prostate-specific antigen (PSA) test, which measures the amount of PSA in the blood, a marker of potential PCa [2]. However, PSA levels may also arise due to other conditions, including an enlarged or inflamed prostate [3]. Therefore, if the PSA test is positive, the patient typically undergoes a multiparametric magnetic resonance imaging (mpMRI) examination [2]. Here, T2-weighted (T2w) and diffusion-weighted (DWI) images (along with apparent diffusion coefficient (ADC) maps) are acquired to investigate the anatomy and detect the presence of the tumor, respectively. These acquisitions allow radiologists to make a preliminary diagnosis following the Prostate Imaging Reporting and Data System (PI-RADS) guidelines [4]. According to the PI-RADS standard, the radiologist assigns a numerical value between 1 and 5 to the suspected lesion: the higher the score, the greater the likelihood that the accounted nodule is malignant. If PI-RADS ≥ 3, the lesion is likely to be clinically significant, and the patient undergoes a biopsy [2]. Based on the two most common patterns in the biopsy specimen, the pathologist assigns a score known as the Gleason Score (GS) to the tumor’s aggressiveness. Along with the GS, it is also recommended to provide the group affiliation of the assigned score defined by the International Society of Urological Pathology (ISUP), as this facilitates predicting patient outcomes and patient communication [5]. If GS ≥ 3+4 (ISUP ≥ 2), the tumor is confirmed to be clinically significant [6]. However, it is often the case that the suspected lesion results are indolent after a biopsy examination. In particular, only about 20% of all PI-RADS 3 patients biopsied show intermediate/severe PCa pathology [7]. Although mpMRI investigation reduces overdiagnosis [8], it remains a qualitative diagnostic tool, highly dependent on radiologist experience and acquisition protocols [9]. For this reason, there is a need for an automated tool to support radiologists in the clinical practice to make diagnosis more robust, reliable, and, above all, non-invasive.

To date, several studies aimed to build machine and deep learning models for the automatic classification of PCa from mpMRI images [10], exploring various techniques, including utilizing generative methods [11], which currently represent the forefront of performance enhancement in this field. Most of these distinguish clinically significant from non-significant PCa (GS ≤ 3 + 3, ISUP ≤ 1 vs. GS ≥ 3 + 4, ISUP ≥ 2) [12,13,14,15,16,17,18]. However, an even more critical task is to distinguish lesions based on their aggressiveness, i.e., low-grade (LG) (GS ≤ 3 + 4, ISUP ≤ 2) vs. high-grade (HG) (GS ≥ 4 + 3, ISUP ≥ 3) lesions, as this is what discriminates the patient’s clinical path. Indeed, all patients with GS ≤ 3 + 4 (ISUP ≤ 2) typically undergo active surveillance [19], even though a lesion with GS = 3 + 4 is still clinically significant. In [20], the radiomic approach was exploited by classifying features extracted from mpMRI images employing a k-nearest neighbor (KNN) algorithm. In [21], the authors employed AlexNet according to a transfer-learning approach by fine-tuning the last classification layer with T2w axial, T2w sagittal images, and ADC maps jointly. In [22], the authors explored both radiomic and deep learning approaches, training several classical machine learning algorithms and 2D convolutional neural networks (CNNs) (with and without attention gates [23]) exploiting T2w axial images only, ADC maps only, and the combination of the two modalities.

In this work, we propose to perform the PCa aggressiveness classification task from T2w images by exploiting an ensemble of vision transformers(ViTs) [24]. ViTs are becoming increasingly popular in the medical imaging domain [25,26,27,28,29,30,31,32,33,34], usually outperforming classical CNNs [35,36], which are one of the most significant networks in the deep learning field [37]. The existing literature typically employs ViTs in transfer learning scenarios by pre-training them on large datasets of natural images and fine-tuning them on specific datasets [27,28,38]. However, due to the limited availability of medical imaging data, we propose training the ViT from scratch by downsizing the architecture. Additionally, since medical data is often acquired in volumetric form, we modify the ViT’s architecture to train it on 3D volumes, leveraging most of the information from the acquisitions. To further enhance the performance of vanilla 3D ViTs, which we will call *base* models henceforth, we propose to combine them in stacking ensembles. The aim is to create a meta-model that learns how to best combine the predictions of base 3D ViTs to harness the capabilities of a stack of models. Finally, to assess the models’ confidence in making predictions in addition to the sole accuracy, we propose two confidence metrics based on the models’ output probability.

The key contributions of our work can be summarized as follows:We introduce a downscaled version of the ViT architecture and train it from scratch using small datasets, addressing the challenge of limited data availability.We propose modifications to the ViT architecture to handle 3D input, enabling the model to effectively leverage volumetric data in the PCa aggressiveness classification task from T2w images.We develop stacking ensembles by combining multiple base 3D ViTs, thereby leveraging the strengths of both stronger and weaker base models to improve overall performance.We define two novel confidence metrics that provide insights into the models’ confidence in making predictions, offering a more comprehensive evaluation beyond accuracy alone.We conduct comparative experiments to assess the performance of ensemble 3D ViTs against the base models in the task of PCa aggressiveness assessment from T2w images.

These contributions collectively aim to enhance the classification accuracy and reliability of PCa aggressiveness assessment, utilizing the power of ensemble models and tailored adaptations to the ViT architecture.

## 2. Dataset

The dataset exploited for this study was collected at Radboud University Medical Centre’s Prostate MR Reference Center as part of the ProstateX-2 challenge [39] and contained a retrospective set of prostate MR studies. All studies included T2w, proton-density-weighted (PDw), dynamic contrast-enhanced (DCE), and DWI acquisitions, along with ADC maps. All images were acquired on two Siemens 3T MR scanners, the MAGNETOM Trio and Skyra. In this work, we exploited only axial T2w acquisitions since, according to [22], they are the most informative for this task. T2w images were acquired using a turbo spin echo sequence with a resolution of around 0.5 mm in the plane and a slice thickness of 3.6 mm.

The dataset contained 112 lesions from 99 male patients (age range: 42–78, age mean and standard deviation (SD): 65 ± 6), subdivided as follows: 77 LG (69%) and 35 HG (31%) with respect to the tumor’s aggressiveness and 50 peripheral (PZ) (44%), 47 anterior fibromuscular stroma (AS) (43%), and 15 transition (TZ) (13%) with respect to the lesion location.

## 3. Methods

### 3.1. Base 3D ViTs

The ViT model was originally designed for handling two-dimensional data, as introduced in [24]. We modified the model to handle three-dimensional input, i.e., each embedding is obtained by flattening a 3D patch rather than a 2D one. Following the formalism presented in [24], we define our input as $x\in R^{H\times W\times Z\times C}$, where (H,W,Z) represents the resolution of the volumetric input, and *C* denotes the number of channels. The ViT divides the input volume into (P,P,Z) patches and flattens them into a one-dimensional vector. As a result, the encoder receives a sequence of flattened patches $x_{p}\in R^{Nx(P^{2}\cdot Z\cdot C)}$ as input for each input volume, where $N=HW/P^{2}$ represents the number of patches. In this study, we deal with grayscale images, so C=1.

Initially, we attempted to apply transfer learning by fine-tuning ViTs based on the positive results reported in the literature [27,28,38]. However, this approach resulted in poor classification performance. We attributed this outcome to the limited size of our training set. Consequently, we explored training the ViTs from scratch to mitigate the issue of overfitting. Thus, we downscaled the architecture, significantly reducing the number of learnable parameters. To determine the optimal configuration, we conducted a grid search, exploring different values for the multilayer perceptron (MLP) size (*d*), hidden size (*D*), number of layers (*L*), and number of attention heads (*k*). We set P=16 as the patch size, which seemed reasonable for the 3D ViTs to process sufficient information. We also conducted preliminary experiments with patch sizes of P=8 and P=32 but observed notably inferior results. After setting the values for *L* and *k*, we derived the appropriate value for *D* based on Equation (Equation 1)
(1)$$D=\frac{P^{2}C}{k}$$
Finally, we calculated *d* value according to Equation (Equation 2):(2)$$d=P^{2}CN$$
In our grid search, we also took into consideration the value of *d* used in the ViT-base architecture proposed in the original article, which was set to 3072 [24]. The parameters of each configuration explored in the grid search are summarized in Table 1. Furthermore, Figure 1 provides a visualization of a generic base 3D ViT architecture, highlighting the parameters that were varied during the grid search.

### 3.2. Ensemble 3D ViTs

Once we defined and trained the 18 base models according to the grid search, we explored combining these models to improve overall performance. Specifically, we implemented several stacking ensembles by concatenating the outputs of the base models and feeding them as input to an additional meta-classifier. This meta-classifier is responsible for producing the final output by performing a linear transformation of the incoming data, as depicted in the following equation:(3)$$y=xA^{T}+b,$$
where, *x* represents the input data, $A^{T}$ denotes the transpose of the weight matrix *A*, and *b* represents the bias term. This process yields the final output, denoted by *y*, which serves as the ensemble model’s prediction based on the combined knowledge from all the base models. Figure 2 provides a visual representation of a generic stacking ensemble consisting of *m* base 3D ViTs. In our study, we evaluated all possible combinations for m=2 and m=3. For each ensemble model, we passed the output of the meta-classifier through a sigmoid function, which computed the probability of the input belonging to the positive class. We considered a prediction as positive if the output probability exceeded 0.5.

### 3.3. Data Pre-Processing

To make the model focus on the tumor lesions, we adopted a slice selection strategy based on the lesion coordinates within the volume. For each volumetric acquisition, we chose the slice containing the lesion itself, as well as two slices above and two slices below it, totaling five slices per lesion. The rationale behind this approach was to provide the network with a section of the entire acquisition that is most influenced by the presence of the tumor lesion. By including the surrounding slices, we aimed to capture contextual information and provide the model with a more comprehensive view of the lesion and its immediate surroundings.

To address the issue of varying matrix sizes in the dataset (in terms of the number of pixels in rows and columns), we adopted a resampling approach. In particular, we selected the largest and most common matrix dimension as our reference size, which was 384×384 in this case, and we upsampled all the images, utilizing bilinear interpolation to match the reference size. This resampling technique allowed us to standardize the image dimensions across the dataset. The decision to choose the largest matrix dimension as the reference in our study was driven by two main considerations. Firstly, by selecting the largest matrix dimension, we aimed to minimize the need for down-sampling, thereby avoiding potential loss of valuable information that may occur during this process. Secondly, this choice was aligned with the most common size found in the dataset, reducing the number of patients that would require resampling. Another crucial aspect was that all acquisitions in the dataset shared the same slice thickness of 3 mm, eliminating the need for any modifications.

After upsampling the images to a consistent size of 384×384 pixels, we performed a center-cropping operation on each slice to facilitate the model’s focus on the prostate gland. The center-cropping process involved extracting a smaller region from the center of each slice. Through empirical analysis, we determined that a crop size of 128×128 pixels was suitable for encompassing prostate glands of various sizes within the field of view while eliminating a significant portion of non-glandular tissues. Thus, for each lesion, we obtained a volume of 128×128×5 pixels, representing the cropped slices from the selected region. In Figure 3, we provide an example of an image during the intermediate steps of upsampling and cropping.

To address the class imbalance issue in the dataset, we applied several data augmentation techniques to balance the training dataset. Specifically, we utilized three augmentation strategies: vertical flip, horizontal flip, and rotation. We chose these strategies to perform rigid transformations of the images while preserving the appearance and shape of the lesion(s) and prostate within each image. During rotation augmentation, we randomly selected the degree of rotation from the interval of [−25°, +25°]. Importantly, we rotated all images within the same volume by the same amount to maintain consistency. Bilinear interpolation was used during the rotation process to ensure smooth transitions. Since the original training set consisted of 54 LG and 27 HG cases, we randomly selected 9 HG volumes using a fixed seed. We generated three augmented versions of each selected HG volume using the techniques described above. Consequently, the final training set comprised 54 volumes of both LG and HG cases, resulting in a balanced dataset. To ensure data harmonization across the training, validation, and test sets, we applied mean normalization by calculating the average pixel value of the training set and subtracting it from all images in the training, validation, and test sets. This normalization step helped to standardize the pixel values across the different sets and align their distributions.

### 3.4. Metrics

#### 3.4.1. Accuracy Metrics

To assess the models’ classification performance, we evaluated the following: specificity, sensitivity, balanced accuracy, the area under the receiving operating curve AUROC, and the area under the precision–recall curve (AUPRC). We chose to report all these metrics because each provides valuable insights into the classifier’s performance from different perspectives. Together, they offer a comprehensive evaluation of the model’s ability to discriminate between the two classes and help in understanding its strengths and limitations. Here’s why each metric is important:**Specificity:** Specificity measures the proportion of true negatives correctly identified by the classifier. It indicates how well the model can correctly identify negative instances, avoiding false positives.**Sensitivity:** Sensitivity measures the proportion of true positives correctly identified by the classifier. It represents the ability of the model to correctly identify positive instances, avoiding false negatives.**Balanced Accuracy:** Balanced accuracy is the arithmetic mean of sensitivity and specificity. A metric like this can be useful in this case since the dataset is imbalanced and traditional accuracy may be unreliable.**AUROC:** AUROC summarizes the overall performance of the model with a single value that indicates the model’s ability to distinguish between the two classes across all possible thresholds.**AUPRC:** AUPRC quantifies the overall ability of the model to balance precision and recall across all possible thresholds. The reason we reported this metric is that our dataset has a moderate skew toward the negative. Therefore, we could better assess our models’ performance considering their behavior with respect to the positive class, regardless of the composition of the dataset itself [40].

#### 3.4.2. Confidence and Calibration Metrics

In contrast to the common practice in the literature where model performance is primarily evaluated based on accuracy metrics, we recognized the importance of considering the models’ prediction confidence. This is crucial, especially in high-risk domains, such as medicine, where the impact of incorrect predictions can be significant. To assess the models’ confidence in making predictions, we introduced two metrics based on the models’ output probabilities. Specifically, we defined a reliable prediction for the negative class as one that is correct and computed with a probability of less than 0.3. Conversely, for the positive class, a reliable prediction is accurate and performed with a probability greater than 0.7. In terms of notation, let us denote true positives as TP, true negatives as TN, false positives as FP, and false negatives as FN. The confidence metrics can be defined as follows: (4)$$Confident\;Specificity\;\left(CSP\right)=\frac{{TN|}_{Probability\;<\;0.3}}{TN+FP},$$
(5)$$Confident\;Sensitivity\;\left(CSE\right)=\frac{{TP|}_{Probability\;>\;0.7}}{TP+FN}.$$
Ideally, *CSP* equals to specificity, and *CSE* equals to sensitivity. When this happens, all the correct predictions are made confidentially, so for negative ground truth (*LG* in this case) instances, the model predicted that the input image had a probability of belonging to the positive class less than or equal to 0.3. On the other hand, for positive ground truth (*HG* in this case) instances, the model predicted that the image had a probability of belonging to the positive class greater than or equal to 0.7. *CSP* and *CSE* are hybrid metrics that combine the ability of the model to make a correct prediction with its confidence in making that prediction. They thus provide a truer and more reliable measure of the model’s potential, which is critical in high-risk domains such as medical imaging. By incorporating these confidence-based metrics into our evaluation process, we sought to provide a more nuanced understanding of the models’ performance, allowing a more cautious and informed approach to their practical application.

Indeed, the reliability of the confidence metrics depends on the calibration of the model. Calibration ensures that the predicted probabilities accurately represent the true probabilities of correctness. In a well-calibrated model, if the model assigns a probability of 40% to an image representing a dog, the actual probability of correctness should be close to 40%.

To assess the calibration of our model, we employed the Brier score (*BS*) [41], a widely used metric for evaluating calibration. The Brier score measures the mean squared difference between the predicted probabilities and the corresponding true outcomes. Mathematically, the Brier score is defined as follows:(6)$$BS=\frac{1}{N}\sum_{i=1}^{N}{\left(f_{i}-o_{i}\right)}^{2}.$$
Here, $f_{i}$ represents the predicted probability, $o_{i}$ denotes the actual outcome for instance *i*, and *N* represents the total number of predictions. In a binary classification task, a perfectly calibrated model would yield a Brier score of 0. This implies that the model consistently assigns a probability of 0 to the negative class and a probability of 1 to the positive class. By computing the Brier score separately for each class, namely the negative class (*BSNC*) and the positive class (*BSPC*), we can assess the calibration of the model for each class individually. Lower values of *BSNC* and *BSPC* indicate better calibration for the negative and positive classes, respectively. The mathematical definition of *BSNC* and *BSPC* is the following:(7)$$BSNC=\frac{1}{N}\sum_{i=1}^{N}{\left(f_{i}^{0}-0\right)}^{2}=\frac{1}{N}\sum_{i=1}^{N}{\left(f_{i}^{0}\right)}^{2},$$
(8)$$BSPC=\frac{1}{N}\sum_{i=1}^{N}{\left(f_{i}^{1}-1\right)}^{2}.$$
Here, $f_{i}^{0}$ represents the predicted probability when the ground truth is negative, and $f_{i}^{1}$ represents the predicted probability when the ground truth is positive. Lower values of *BSNC* indicate better calibration for the negative class, while lower values of *BSPC* indicate better calibration for the positive class.

## 4. Experiments

### 4.1. Dataset Splitting

In our study, we initially divided the dataset into two parts: 90 lesions (80%) for training and validation and 22 lesions (20%) for the test set. To ensure robustness in model evaluation, we employed two different strategies for further splitting the 90-lesion set. The first strategy involved creating two subsets: 90% for training and 10% for validation. The second strategy involved dividing it into five subsets for conducting a five-fold cross-validation (CV). We used the first splitting strategy to train the 18 base 3D ViTs, which we later used in the ensembles. We employed the second splitting strategy for optimizing both the base and ensemble models. To enhance the reliability of our findings, we bootstrapped the training set obtained from the first splitting strategy, generating 100 samples. We utilized these samples for re-training the best-performing base and ensemble models obtained from the five-fold CV. To ensure balanced representation, we performed all splits while stratifying the data based on the lesion class (approximately two-thirds LG and one-third HG) and lesion location (approximately two-fifths PZ, two-fifths AS, and one-fifth TZ). Additionally, we employed a patient-wise splitting approach to prevent any data leakage. A summary of the composition of the training, validation, and test sets with respect to tumour aggressiveness and location can be found in Table 2.

### 4.2. Training Setup

Training, validation, and test phases, both for base and ensemble models, were coded in Python by employing the following modules: Pytorch (v. cuda-1.10.0) [42], Numpy (v.1.20.3) [43], Scikit-learn (v. 0.24.2) [44], Pydicom (v. 2.1.2) [45], Pillow (v. 9.0.1) [46], and Pandas [47]. We performed training and test processes employing an Intel Core i7 ASUS Desktop Computer with 32 GB RAM and an NVIDIA GeForce GTX 1650 GPU.

#### 4.2.1. Base Models

We trained each of the 18 base models with the following hyperparameters: learning rate = 1 × 10${}^{-4}$, weight decay = 1 × 10${}^{-2}$, maximum number of steps = 1000, batch size = 4, warmup steps = 1000, optimization algorithm = Adam, loss function = BinaryCrossEntropy. To make each training reproducible, we exploited the reproducibility flags provided by Pytorch [42], Numpy [43], and Random [48] libraries, choosing a seed equal to 42. We trained each base model according to a five-fold CV.

#### 4.2.2. Ensemble Models

We explored all possible combinations of two- and three-base-model stacking ensembles. Specifically, we evaluated 153 two-model combinations and 816 three-model combinations. Each combination underwent training and evaluation using a five-fold CV approach, maintaining consistency with the dataset splitting and reproducibility seed used for the base models. To train the meta-classifier, we utilized the following set of hyperparameters: learing rate = 1 × 10${}^{-4}$, weight decay = 1 × 10${}^{-2}$, number of epochs = 100, batch size = 4, optimization algorithm = Adam, loss function = BinaryCrossEntropy. These hyperparameters were applied consistently across all training iterations of the meta-classifier to ensure fair and comparable performance evaluation.

### 4.3. Performance Evaluation

To assess the performance and calibration of the models, we evaluated their accuracy, confidence in predictions, and calibration. The selection of the best-performing base and ensemble model was based on the performance on the five-fold CV process. To compare the performance of the best-performing base and ensemble models, we conducted a 100-sample bootstrap of the entire training set. For each bootstrapped sample, we re-trained both the best-performing base model and the ensemble model. Subsequently, we evaluated each re-trained model on the same hold-out test set generating a performance distribution. From these distributions, we calculated the median performance and the 95% confidence interval (CI). This approach allowed us to obtain a robust estimation of the models’ performance and CI for a reliable performance assessment and comparison.

#### Statistical Analysis

To evaluate the statistical significance of the difference between the best-performing base and ensemble models, we conducted the Wilcoxon signed-rank test. We considered the difference between the performance distributions of the base and ensemble models statistically significant if the resulting *p*-value (*p*) was less than 0.050. We performed the statistical analysis exploiting the Scipy library [49] (v. 1.9.3).

### 4.4. Results

The best-performing ensemble of 3D ViTs, composed of configurations 5, 9, and 11, achieved the following results in terms on median and 95% CI: a specificity of 0.83 [0.67–1], sensitivity of 0.67 [0.67–1], balanced accuracy of 0.75 [0.67–0.96], AUROC of 0.89 [0.64–1], and AUPRC of 0.87 [0.57–1]. In terms of confidence prediction, the ensemble yielded the following results: a CSP of 0.83 [0.41–1], CSE of 0.33 [0–1], a BSNC of 0.10 [0–0.25], a BSPC of 0.31 [0.02–0.41], and an overall BS of 0.16 [0.08–0.24]. For a random classifier, the AUROC and AUPRC values on this test set were 0.500 and 0.273, respectively.

Regarding the best-performing base 3D ViT with configuration 5, it achieved the following results on the external test set in terms of median and 95% CI: a specificity of 0.75 [0.75–0.75], sensitivity of 0.83 [0.67–0.83], balanced accuracy of 0.79 [0.71–0.79], AUROC of 0.86 [0.85–0.89], and AUPRC of 0.65 [0.60–0.68]. In terms of confidence prediction, the base model yielded the following results: CSP of 0.68 [0.63–0.69], a CSE of 0.50 [0.50–0.59], a BSNC of 0.15 [0.14–0.16], a BSPC of 0.12 [0.11–0.17], and an overall BS of 0.14 [0.14–0.15].

We presented the results for the best-performing base and ensemble models in terms of accuracy and confidence in Table 3 and Table 4, respectively. Additionally, in Figure 4, we included the ROC and PR curves for the two best-performing models when trained on the entire training set (non-bootstrapped). These visualizations provide further insights into the performance and discriminative capabilities of the models.

After conducting the Shapiro–Wilk test, we determined that all the distributions deviated from normality with *p* < 0.001. Consequently, we employed the Wilcoxon signed-rank test to assess the statistical differences between the base and ensemble models. The test yielded the following statistics for each metric: W=1323.0, p<0.001 for specificity; W=91.0, p<0.001 for sensitivity; W=1368.0, p<0.001 for balanced accuracy; W=2425.0, p=0.73 for AUROC; W=159.0, p<0.001 for AUPRC; W=1246.0, p<0.001 for CSP; W=812.5, p<0.001 for CSE; W=911.0, p<0.001 for BSNC; W=122.0, p<0.001 for BSPC; and W=1427, p<0.001 for BS.

In Figure 5, we present the box plots illustrating the distributions of results for the best-performing base and ensemble models based on accuracy metrics. Similarly, in Figure 6, we provide the box plots representing the distributions of results for the base and ensemble models with respect to confidence metrics.

For clarity and comparison purposes, we present the five-fold CV accuracy and confidence results for all the base models in Table 5 and Table 6, respectively. Similarly, we provide the five-fold CV results of the top 10 best-performing ensembles in terms of accuracy and prediction confidence in Table 7 and Table 8, respectively. All reported values are presented as the mean and SD across the five folds.

## 5. Discussion

In this study, we proposed a trained-from-scratch 3D ViT in a stacking ensemble configuration and assessed its effectiveness in assessing PCa aggressiveness from T2w MRI acquisitions. The approach employed in this study involved training vanilla 3D ViT in various configurations and subsequently combining them into 2- and 3-model ensembles. We concatenated the features extracted from each of these base models and provided it to a single fully-connected layer, which yielded the final prediction. The evaluation of the trained models centred around assessing their accuracy performance, specifically focusing on measures such as specificity, sensitivity, balanced accuracy, AUROC, and AUPRC. Additionally, we measured the calibration of these models using BS. We further computed BS with respect to the negative class only BSNC and the positive class only (BSPC), allowing us to gain insights into their calibration for each class separately. To enhance the model evaluation process, we proposed the introduction of two novel metrics, CSP and CSE. The primary purpose of these metrics was to provide a comprehensive measure that combined the model’s prediction capabilities with its confidence level. By doing so, we aimed to highlight only those predictions made with a confidence level above a predefined threshold. The implementation of CSP and CSE aimed to offer more detailed and reliable information about the model’s performance, with the ultimate goal of bringing its practical application in clinical settings closer. These metrics were designed to provide valuable insights into the accuracy and confidence of the model’s predictions, enabling a more informed and cautious approach when utilizing the model’s outputs in real-world medical scenarios.

We trained the base 3D ViT models according to a grid search, varying architecture parameters such as *d*, *D*, *L*, and *k*, for a total of 18 base 3D ViTs. By training all possible combinations of two and three models, we created 966 ensembles from these base models. We trained each ensemble following a five-fold CV and re-trained the one with the highest performance on a 100-sample bootstrapped training set. For comparison, we optimized the 18 base ViTs as well using a five-fold CV and re-trained the best-performing base ViT on the same bootstrapped training set. We evaluated each of the 100 models from both the ensemble and base ViT on a separate hold-out test set. To determine whether there was a significant difference in performance between the base and ensemble models, we conducted a statistical analysis on the distributions of results obtained from the test set evaluations.

We evaluated our approach using the ProstateX-2 challenge dataset [39] appropriately divided into training, validation and test set, ensuring strict separation between patients.

According to the results, the ensemble model demonstrated strong performance in classifying LG and HG lesions, as evidenced by its high median AUROC (0.89). A high AUROC indicates a robust ability to accurately identify positive instances while effectively minimizing false positive predictions. This performance metric holds great importance, particularly in tasks characterized by imbalanced class distributions or situations where the costs associated with false positives and false negatives are substantial, as exemplified in this case. The model also demonstrated good results in classifying the positive class specifically, with a median AUPRC of 0.87. Apart from its general performance, the model displayed notable proficiency in accurately classifying the positive class. The median AUPRC was recorded at a median value of 0.87. The AUPRC is a crucial evaluation metric, especially when dealing with imbalanced datasets. To elaborate further, the AUPRC measures the area under the precision–recall curve, which plots the precision against the recall (sensitivity) at various classification thresholds. In cases where the class distribution is imbalanced, a high AUPRC becomes crucial as it signifies that the model effectively achieves a high precision rate while maintaining a reasonable recall rate. This means that when the model makes a positive prediction, it is highly likely to be correct (high precision), and it successfully captures a significant portion of the actual positive instances (high sensitivity).

Regarding the calibration aspect, the model exhibited strong calibration performance, as evidenced by a BS of 0.16. The BS is a widely used metric that assesses the calibration of probabilistic predictions made by a classification model by measuring the mean squared difference between the predicted probabilities and the actual binary outcomes. A lower Brier score indicates better calibration, implying that the model’s predicted probabilities align closely with the actual outcomes.

Concerning the model’s confidence in its predictions, CSP revealed a remarkable value equal to the classical specificity metric, reaching 0.83. This outcome signifies that all correct predictions related to the negative class (specificity) were accompanied by high confidence levels, i.e., the model returned one with an output probability less than or equal to 0.3.

Upon comparing the ensemble model to the best-performing base 3D ViT, we conducted a Wilcoxon signed-rank test to assess the statistical significance of their performance differences. In terms of AUROC, the results of this analysis revealed no statistically significant difference between the two models (*p* = 0.73). This indicates that both the ensemble model and the base ViT performed similarly in terms of overall discriminative ability. However, a notable contrast emerged when evaluating the models’ proficiency in classifying HG lesions. Indeed, the ensemble model outperformed the base ViT by a 22% improvement in AUPRC, which resulted in statistically significant (*p* < 0.001). The substantial improvement in AUPRC for HG lesions highlights the ensemble model’s particular strength in accurately identifying and distinguishing severe lesions from the base ViT. This is crucial in medical applications where detecting HG lesions can significantly impact patient outcomes and treatment decisions.

Indeed, while the ensemble model exhibited a significant performance improvement in classifying HG lesions, it is essential to consider its impact on calibration and confidence in predictions. A closer examination of the calibration metrics reveals that the ensemble model’s performance comes at the expense of poorer calibration towards the HG class. This is evident from the higher BSPC and the lower CSE values compared to the base ViT. The higher BSPC suggests that the ensemble model’s probability predictions for positive instances in the HG class may be less well-calibrated. Consequently, this may lead to less reliable probability estimates for high-grade lesions. Furthermore, the lower CSE value indicates reduced confidence in the ensemble model’s predictions for the HG class, i.e., the model might not be as certain when making predictions for positive instances in this category. On the contrary, the ensemble model demonstrated improved confidence in its predictions for the negative class when compared to the base ViT. This was evident from the lower BSNC and CSP values relative to the base model. The lower BSNC suggests that the ensemble model’s probability predictions for the negative class are better calibrated and align closely with the actual outcomes. Moreover, the lower CSP metric signifies that the ensemble model confidently assigns lower probabilities (less than or equal to 0.3) to the correct negative predictions. All differences in cited metrics values resulted in statistically significant according to the Wilcoxon signed-rank test (*p* < 0.001).

Furthermore, we investigated the performance of the ensemble model compared to the base model focusing on the consistency of its predictions. The results, illustrated in Figure 5 and Figure 6, showcased the ensemble model displayed larger CIs with respect to to the base model. This suggests that the ensemble model’s performance is characterized by higher variability, making its predictions less stable and more susceptible to fluctuations. While the ensemble model demonstrated superior performance in certain aspects, the broader range of its CIs indicates that its predictions may be less consistent, leading to varying results across different iterations or datasets. This variability in performance could be attributed to the complexity introduced by the ensemble configuration, as it involves combining multiple models, each with its unique characteristics.

**Performance of best-performing base and ensemble models.** The best-performing ensemble model contains the best-performing base model, indicating that this specific configuration of architecture parameters achieves high performance. This configuration stands out as it is present in three out of the ten best ensemble combinations, as shown in Table 7.**Impact of the number attention heads and embedding size.** Configurations 5 and 8, both having eight attention heads and an embedding size of 32, are the most frequently recurring setups in the top-performing ensembles (three out of ten combinations each). Additionally, configurations 4, 7, and 16, all having an embedding size of 64 and four attention heads, also occur frequently (three out of ten ensembles). These architecture parameter combinations seem to contribute to improving the ensembles’ performance, even though they might not perform well when used in isolation.**Influence of MLP size.** The size of the MLP (represented by the parameter *d*) does not significantly impact performance. The results indicate that there is no noticeable difference in performance between models trained with *d* = 2048 or *d* = 3072, whether in the base or ensemble models. This suggests that increasing the MLP size beyond a certain point does not lead to significant improvements in accuracy.**Stacking ensemble effect.** Comparing Table 5 and Table 7, it is evident that combining weaker models in stacking ensembles significantly improves accuracy performance. This finding highlights the benefits of leveraging ensemble methods to improve model performance, even when individual base models might not be as strong.

We compared the performance of our best-performing ensemble 3D ViT model with state-of-the-art (SoTA) studies that addressed the same clinical task of assessing PCa aggressiveness from MRI images. To ensure a fair comparison, we considered all the studies that employ T2w images alone or combined with other images modalities. In addition, we compared our results with studies employing both radiomics and deep learning for a comprehensive evaluation. In radiomic studies, Jensen et al. [20] achieved an AUROC of 0.830 using a K-nearest neighbors (KNN) classifier with only T2w images. Bertelli et al. [22] reported the best machine learning model with an AUROC of 0.750 (90% CI: [0.500–1.000]) for T2w images alone and 0.63 (90% CI: [0.17–1]) when combining T2w and ADC images. As for deep learning models, Yuan et al. [21] used an AlexNet in a transfer-learning approach, achieving an AUROC of 0.809 with T2w images alone and 0.90 when combining T2w and ADC acquisitions. Bertelli et al. [22] utilized CNNs with attention gates, resulting in an AUROC of 0.875 (90% CI: [0.639–1.000]) for T2w images only and 0.67 (90% CI: [0.30–1]) when combining T2w and ADC. In comparison, our best-performing ensemble 3D ViT model displayed SoTA performance with an AUROC that outperformed all the other models when using only T2w images. Unfortunately, a direct comparison for the positive class only was not possible due to the lack of available data in the literature. The comparison results are summarized in Table 9, providing an overview of our model’s performance concerning the SoTA studies.

This study is subject to several limitations that should be acknowledged. Firstly, we did not conduct a hyperparameter optimization, which may have limited the overall performance of our models. Instead, we kept the hyperparameter values fixed throughout all the training phases and focused solely on optimizing the architecture parameters. Moreover, in our investigation of stacking ensembles, we only considered two- and three-model combinations. Expanding the ensemble to include more base models might lead to further improvements in the results. Indeed, utilizing a larger ensemble could enhance model diversity and potentially increase predictive performance. Furthermore, one of the limitations lies in the choice of using only axial T2w images for training. In contrast, many of the cited works combined multiple modalities, such as T2w and ADC images, to improve model performance. Expanding the dataset to incorporate additional modalities could potentially enhance the model’s ability to capture diverse and complementary information, thereby leading to more robust predictions. Overall, recognizing these limitations is essential to interpreting the findings accurately and understanding the scope of the study. Future research could address these limitations and explore new avenues for improving the performance of the 3D ViT model in assessing PCa aggressiveness.

## 6. Conclusions

In this study, we explored the effectiveness of 3D ViTs in stacking ensembles to improve the assessment of PCa aggressiveness from T2w images. The best-performing ensemble demonstrated strong capabilities in differentiating between LG and HG lesions, achieving an AUROC comparable to the SoTA methods. Additionally, it displayed a high ability to classify positive instances, as evidenced by a high AUPRC, also outperforming the base model (*p* < 0.001). However, this improvement came at the cost of reduced calibration and prediction confidence, as indicated by a lower CSE and higher BSPC. At the same time, however, the ensemble model was found to be better calibrated and more confident in its predictions regarding the negative class. In addition, the larger CI for the ensemble model suggests that its performance is less reliable and more variable compared to the base model. Overall, our study showed that 3D ViT ensembles yield promising results for PCa aggressiveness assessment from T2w images and provide improved even though less reliable performance with respect to their base version. This suggests that further refinements or strategies to mitigate variability may be necessary before its deployment in critical real-world scenarios.

Our analysis has certain limitations. First, we did not perform hyperparameter optimization. Secondly, we limited our investigation to ensembles of only 2 and 3 models. Finally, we trained our models only on T2w images from a relatively small dataset. In fact, at the time of our experiments, ProstateX-2 was one of the few publicly available datasets with more than 100 lesions. Our work was a preliminary investigation that will serve as the basis for further research taking advantage of larger datasets that are being collected within the Tuscany Region PAR FAS NAVIGATOR project and the EU H2020 ProCAncer-I project. In these new research endeavors we expect to address these aspects and thereby improve the effectiveness of our approach.

## Figures and Tables

**Figure 1 bioengineering-10-01015-f001:**
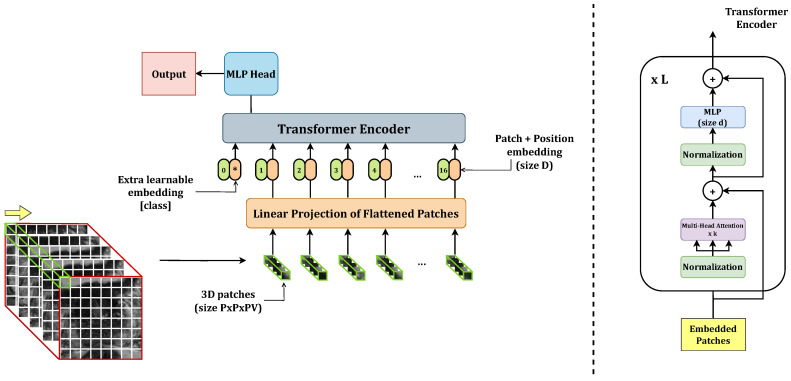
Base 3D ViT.

**Figure 2 bioengineering-10-01015-f002:**
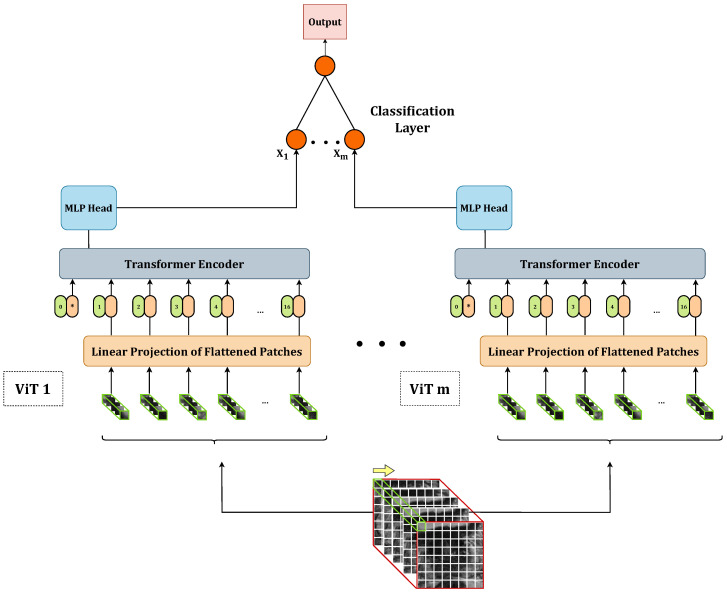
3D ViTs stacking ensemble.

**Figure 3 bioengineering-10-01015-f003:**
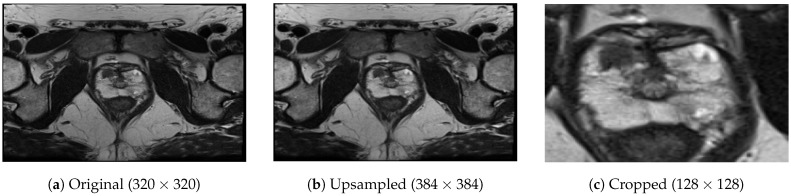
Image examples during intermediate pre-processing steps.

**Figure 4 bioengineering-10-01015-f004:**
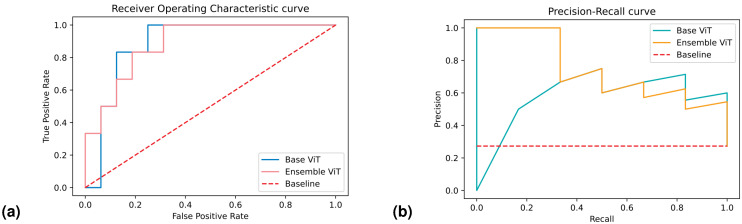
ROC and PR curves of the best-performing base ensemble 3D ViTs trained and the whole training set and evaluated on the hold-out test set.

**Figure 5 bioengineering-10-01015-f005:**
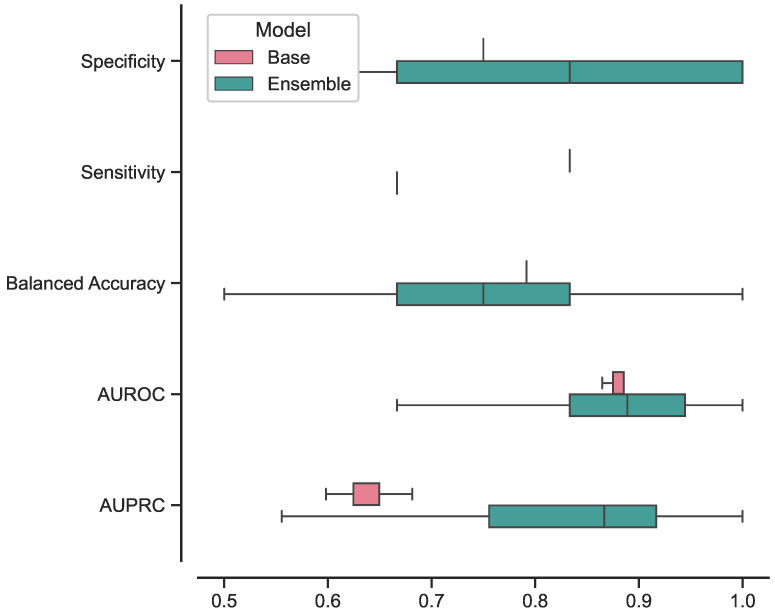
Box plot of accuracy results distributions for best-performing base and ensemble models.

**Figure 6 bioengineering-10-01015-f006:**
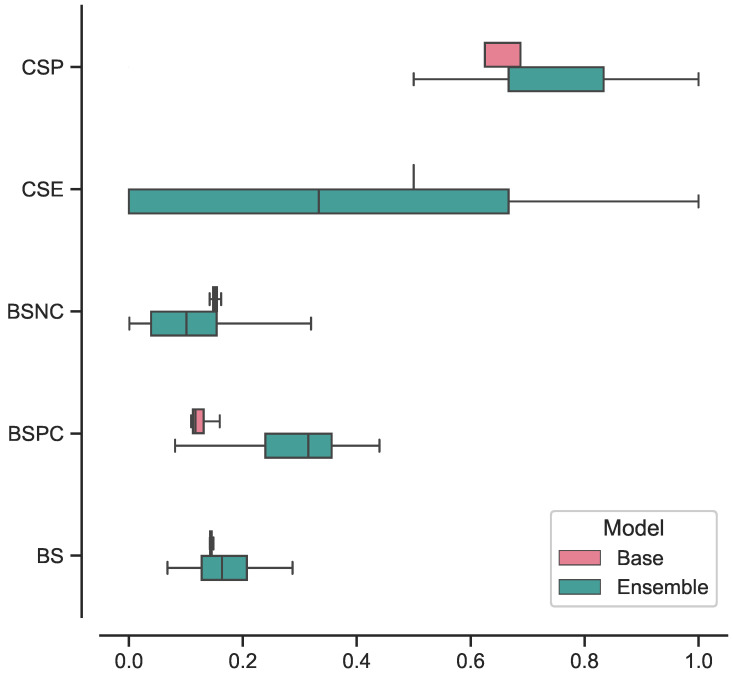
Box plot of confidence results distributions for best-performing base and ensemble models.

**Table 1 bioengineering-10-01015-t001:** Architecture parameters for each base 3D ViT configuration of the grid search.

P	d	L	D	k	Configuration
16	2048	4	64	4	1
			32	8	2
			16	16	3
		6	64	4	4
			32	8	5
			16	16	6
		8	64	4	7
			32	8	8
			16	16	9
	3072	4	64	4	10
			32	8	11
			16	16	12
		6	64	4	13
			32	8	14
			16	16	15
		8	64	4	16
			32	8	17
			16	16	18

**Table 2 bioengineering-10-01015-t002:** Dataset splitting according to the second-way criterion.

Dataset	Severity	Location			Total
		**PZ**	**AS**	**TZ**	
Training	LG	25	21	9	**45**
	HG	11	13	2	**26**
*Subtotal*					**81**
Validation	LG	3	3	0	**6**
	HG	1	1	1	**3**
*Subtotal*					**9**
Test	LG	7	7	2	**16**
	HG	3	2	1	**6**
*Subtotal*					**22**
**Total**		**50**	**47**	**15**	**112**

**Table 3 bioengineering-10-01015-t003:** Best-performing base and ensemble models’ accuracy results on the external test set provided as median and 95% CI.

Model	Specificity	Sensitivity	Balanced Accuracy	AUROC	AUPRC
Base	0.75 [0.75–0.75]	0.83 [0.67–0.83]	0.79 [0.71–0.79]	0.86 [0.85–0.89]	0.65 [0.60–0.68]
Ensemble	0.83 [0.67–1]	0.67 [0.67–1]	0.75 [0.67–0.96]	0.89 [0.64–1]	0.87 [0.57–1]

**Table 4 bioengineering-10-01015-t004:** Best-performing base and ensemble models’ confidence results on the external test set provided as median and 95% CI. We indicate with the ↑ symbol the metrics whose values are better when higher, and with the ↓ symbol, we indicate the metrics whose values are better when smaller.

Model	CSP ↑	CSE ↑	BSNC ↓	BSPC ↓	BS ↓
Base	0.68 [0.63–0.69]	0.50 [0.50–0.59]	0.15 [0.14–0.16]	0.12 [0.11–0.17]	0.14 [0.14–0.15]
Ensemble	0.83 [0.41–1]	0.33 [0–1]	0.10 [0–0.25]	0.31 [0.02–0.41]	0.16 [0.08–0.24]

**Table 5 bioengineering-10-01015-t005:** Prediction accuracy results of each configuration of base 3D ViT on the five-fold CV. Results are provided as mean and SD across the five folds. We use **bold** formatting to highlight the configuration that achieved the best performance and underlined formatting to indicate the configuration with the worst performance. Details on configurations can be viewed in Table 1.

Configuration	Specificity	Sensitivity	Balanced Accuracy	AUROC	AUPRC
1	0.683 (0.086)	0.533 (0.074)	0.642 (0.073)	0.631 (0.086)	0.533 (0.074)
2	0.783 (0.125)	0.467 (0.323)	0.625 (0.124)	0.692 (0.161)	0.626 (0.161)
3	0.700 (0.100)	0.633 (0.067)	0.667 (0.075)	0.719 (0.097)	0.544 (0.127)
4	0.700 (0.041)	0.500 (0.211)	0.600 (0.101)	0.647 (0.112)	0.505 (0.080)
**5**	**0.683 (0.033)**	**0.733 (0.094)**	**0.708 (0.104)**	**0.758 (0.090)**	**0.587 (0.123)**
6	0.567 (0.244)	0.767 (0.226)	0.667 (0.126)	0.706 (0.121)	0.571 (0.155)
7	0.667 (0.105)	0.567 (0.249)	0.617 (0.113)	0.686 (0.039)	0.563 (0.069)
8	0.667 (0.053)	0.667 (0.183)	0.667 (0.102)	0.669 (0.112)	0.544 (0.085)
9	0.700 (0.215)	0.467 (0.323)	0.583 (0.079)	0.611 (0.101)	0.467 (0.055)
10	0.667 (0.053)	0.600 (0.226)	0.633 (0.110)	0.706 (0.068)	0.605 (0.121)
11	0.683 (0.062)	0.633 (0.163)	0.658 (0.110)	0.697 (0.170)	0.587 (0.207)
12	0.400 (0.327)	0.767 (0.200)	0.583 (0.075)	0.675 (0.081)	0.567 (0.070)
13	0.500 (0.183)	0.700 (0.125)	0.600 (0.090)	0.608 (0.139)	0.522 (0.127)
14	0.683 (0.232)	0.700 (0.306)	0.692 (0.131)	0.753 (0.098)	0.621 (0.137)
15	0.717 (0.067)	0.467 (0.194)	0.592 (0.093)	0.586 (0.059)	0.444 (0.077)
16	0.550 (0.277)	0.667 (0.183)	0.608 (0.128)	0.617 (0.097)	0.488 (0.084)
17	0.600 (0.314)	0.733 (0.133)	0.667 (0.095)	0.739 (0.100)	0.609 (0.108)
18	0.783 (0.113)	0.667 (0.000)	0.725 (0.057)	0.753 (0.151)	0.646 (0.210)

**Table 6 bioengineering-10-01015-t006:** Prediction confidence results of each configuration of base 3D ViT on the five-fold CV. Results are provided as mean and SD across the five folds. We indicate with the ↑ symbol the metrics whose values are better when higher, and with the ↓ symbol, we indicate the metrics whose values are better when smaller. We use **bold** formatting to highlight the configuration that achieved the best performance and underlined formatting to indicate the configuration with the worst performance. Details on configurations can be viewed in Table 1.

Configuration	CSP ↑	CSE ↑	BSNC ↓	BSPC ↓	BS ↓
1	0.400 (0.213)	0.300 (0.125)	0.245 (0.049)	0.293 (0.123)	0.261 (0.045)
2	0.283 (0.256)	0.200 (0.194)	0.184 (0.024)	0.337 (0.155)	0.235 (0.042)
3	0.050 (0.100)	0.033 (0.067)	0.236 (0.024)	0.238 (0.016)	0.237 (0.014)
4	0.350 (0.291)	0.200 (0.040)	0.230 (0.040)	0.275 (0.084)	0.245 (0.034)
**5**	**0.150 (0.260)**	**0.033 (0.067)**	**0.217 (0.014)**	**0.330 (0.160)**	**0.255 (0.052)**
6	0.100 (0.097)	0.000 (0.000)	0.241 (0.040)	0.227 (0.037)	0.236 (0.021)
7	0.500 (0.264)	0.333 (0.236)	0.207 (0.054)	0.282 (0.092)	0.232 (0.015)
8	0.017 (0.033)	0.033 (0.067)	0.244 (0.026)	0.217 (0.026)	0.235 (0.017)
9	0.200 (0.319)	0.000 (0.000)	0.212 (0.319)	0.282 (0.083)	0.235 (0.025)
10	0.233 (0.255)	0.167 (0.258)	0.252 (0.030)	0.265 (0.064)	0.256 (0.040)
11	0.283 (0.267)	0.200 (0.245)	0.229 (0.069)	0.227 (0.067)	0.228 (0.065)
12	0.133 (0.194)	0.133 (0.194)	0.279 (0.081)	0.204 (0.064)	0.254 (0.040)
13	0.050 (0.067)	0.133 (0.125)	0.255 (0.027)	0.223 (0.031)	0.244 (0.018)
14	0.333 (0.321)	0.433 (0.403)	0.241 (0.129)	0.230 (0.218)	0.237 (0.068)
15	0.383 (0.215)	0.133 (0.163)	0.208 (0.010)	0.343 (0.127)	0.253 (0.037)
16	0.383 (0.267)	0.300 (0.221)	0.262 (0.068)	0.268 (0.112)	0.264 (0.050)
17	0.233 (0.232)	0.467 (0.356)	0.325 (0.233)	0.184 (0.095)	0.278 (0.127)
18	0.417 (0.167)	0.200 (0.194)	0.189 (0.074)	0.253 (0.055)	0.210 (0.064)

**Table 7 bioengineering-10-01015-t007:** Prediction accuracy results of the top-10 stacking ensembles on the five-fold CV. Results are provided as mean and SD across the five folds. We use **bold** formatting to highlight the configuration that achieved the best performance and underlined formatting to indicate the configuration with the worst performance. Details on configurations can be viewed in Table 1.

Ensemble	Specificity	Sensitivity	Balanced Accuracy	AUROC	AUPRC
2 + 4	0.600 (0.113)	0.600 (0.194)	0.600 (0.138)	0.781 (0.115)	0.630 (0.165)
4 + 15	0.850 (0.122)	0.400 (0.309)	0.625 (0.109)	0.775 (0.032)	0.617 (0.053)
7 + 9	0.617 (0.075)	0.600 (0.226)	0.609 (0.100)	0.744 (0.086)	0.610 (0.070)
4 + 7 + 16	0.850 (0.062)	0.600 (0.170)	0.725 (0.068)	0.828 (0.052)	0.745 (0.087)
5 + 8 + 18	0.767 (0.355)	0.633 (0.267)	0.700 (0.122)	0.817 (0.131)	0.753 (0.161)
**5 + 9 + 11**	**0.833 (0.139)**	**0.700 (0.067)**	**0.767 (0.057)**	**0.839 (0.049)**	**0.778 (0.072)**
5 + 10 + 16	0.833 (0.139)	0.600 (0.226)	0.717 (0.072)	0.783 (0.135)	0.722 (0.183)
7 + 8 + 16	0.750 (0.118)	0.767 (0.133)	0.758 (0.096)	0.800 (0.062)	0.702 (0.096)
8 + 10 + 16	0.817 (0.082)	0.733 (0.082)	0.775 (0.033)	0.783 (0.092)	0.639 (0.142)
8 + 13 + 18	0.733 (0.062)	0.500 (0.211)	0.617 (0.081)	0.694 (0.113)	0.565 (0.142)

**Table 8 bioengineering-10-01015-t008:** Prediction confidence results of the top-10 stacking ensembles on the five-fold CV. Results are provided as mean and SD across the five folds. We indicate with the ↑ symbol the metrics whose values are better when higher, and with the ↓ symbol, we indicate the metrics whose values are better when smaller. We use **bold** formatting to highlight the configuration that achieved the best performance and underlined formatting to indicate the configuration with the worst performance. Details on configurations can be viewed in Table 1.

Ensemble	CSP ↑	CSE ↑	BSNC ↓	BSPC ↓	BS ↓
2 + 4	0.500 (0.085)	0.533 (0.194)	0.131 (0.050)	0.152 (0.071)	0.138 (0.057)
4 + 15	0.550 (0.287)	0.267 (0.271)	0.157 (0.075)	0.397 (0.309)	0.237 (0.067)
7 + 9	0.567 (0.067)	0.267 (0.226)	0.074 (0.027)	0.306 (0.074)	0.151 (0.032)
4 + 7 + 16	0.783 (0.113)	0.433 (0.309)	0.096 (0.042)	0.356 (0.153)	0.182 (0.034)
5 + 8 + 18	0.633 (0.310)	0.400 (0.170)	0.122 (0.120)	0.286 (0.160)	0.177 (0.066)
**5 + 9 + 11**	**0.767 (0.207)**	**0.300 (0.267)**	**0.098 (0.066)**	**0.270 (0.045)**	**0.155 (0.032)**
5 + 10 + 16	0.683 (0.249)	0.367 (0.194)	0.138 (0.092)	0.317 (0.121)	0.198 (0.047)
7 + 8 + 16	0.717 (0.085)	0.267 (0.249)	0.122 (0.036)	0.268 (0.068)	0.171 (0.041)
8 + 10 + 16	0.617 (0.125)	0.200 (0.067)	0.135 (0.047)	0.224 (0.064)	0.164 (0.025)
8 + 13 + 18	0.633 (0.100)	0.333 (0.211)	0.191 (0.080)	0.383 (0.118)	0.255 (0.059)

**Table 9 bioengineering-10-01015-t009:** Comparison between our best-performing 3D ViTs stacking ensemble and several SoTA models that assess the PCa aggressiveness classification (ISUP 1 + 2 vs. rest).

Study	Method	Dataset	Image Modality	AUROC	CI
Bertelli et al. [22]	Feature extraction + AdaBoost	Private dataset	T2w	0.75 [0.50–1]	90%
Bertelli et al. [22]	Feature extraction + XGBoost	Private dataset	T2w + ADC	0.63 [0.17, 1]	90%
Bertelli et al. [22]	Attention CNN	Private dataset	T2w	0.88 [0.64–1]	90%
Bertelli et al. [22]	Attention CNN	Private dataset	T2w + ADC	0.67 [0.30, 1]	90%
Jensen et al. [20]	Feature extraction + KNN	ProstateX	T2w	0.83 [-]	-
Yuan et al. [21]	CNN	ProstateX + Private dataset	T2w	0.81 [-]	-
Yuan et al. [21]	CNN	ProstateX + Private dataset	T2w + ADC	0.90 [-]	-
**Ours**	**3D ViTs Ensemble**	**ProstateX**	**T2w**	**0.89 [0.64–1]**	**95%**

## Data Availability

Data can be found at https://wiki.cancerimagingarchive.net/pages/viewpage.action?pageId=23691656, accessed on 29 June 2023.

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
