# Peer review of "3D-Vision-Transformer Stacking Ensemble for Assessing Prostate Cancer Aggressiveness from T2w Images†"

_bioengineering, 2023, doi:10.3390/bioengineering10091015_

Round 1

Reviewer 1 Report

1. Fig. 1 should be presented in a more scientific style. Now it looks good for a scientific-popular magazine but not for a research journal.

2. It is suggested that further discussions and attributions regarding the proposed method and the findings of the experiments can be appended. Also, the current discussions and attributions should be logically reorganized and refined. I suggest the authors read studies performed by scholars such as R. Ranjbarzadeh et al., s. Jafarzadeh et al., and their groups.

3. It is suggested that the conclusion section can also summarize all the findings and attributions, other than the introductions to the proposed methods.

4. Authors should argue their choice of performance evaluation indicators.

5. Discuss the merits, demerits, and possible future extension of the proposed work in the conclusion section. The following papers are good examples:

https://doi.org/10.1155/2022/5052435

https://doi.org/10.3390/bioengineering10040495

Reviewer 2 Report

This paper applies VIT to assess the aggressiveness of prostate cancer. In the case of small datasets, this paper train from scratch using a downscaled version of the VIT. This paper also develops a stacked ensemble by combining multiple base 3D VITs to take advantage of both stronger and weaker base models to improve overall performance. This paper also defines a measure of model confidence for the assessment of prostate cancer. Here are some suggestions:

1. For the figures in the article, each graph has a lot of blank areas, and the arrangement of the elements in the graph is not very beautiful.

2. The dataset used in the article is not very large, and the test on the 20% dataset is not convincing. After all, it is a medical diagnosis, and you should look for more convincing data sets as much as possible.

3. Why finally trained 18 3D Vision Transformers on T2-weighted axial acquisitions? You should verify that several VIT combinations can show the best performance.

4. Some related newly works and surveys should be introduced and cited. For example:

[1]He, Kelei, et al. "Transformers in medical image analysis: A review." Intelligent Medicine (2022).

[2]Golfe, Alejandro, et al. "ProGleason-GAN: Conditional Progressive Growing GAN for prostatic cancer Gleason Grade patch synthesis." Computer Methods and Programs in Biomedicine (2023): 107695.

[3] Li, Zewen, et al. "A survey of convolutional neural networks: analysis, applications, and prospects." IEEE transactions on neural networks and learning systems (2021).

This paper applies VIT to assess the aggressiveness of prostate cancer. In the case of small datasets, this paper train from scratch using a downscaled version of the VIT. This paper also develops a stacked ensemble by combining multiple base 3D VITs to take advantage of both stronger and weaker base models to improve overall performance. This paper also defines a measure of model confidence for the assessment of prostate cancer. Here are some suggestions:

1. For the figures in the article, each graph has a lot of blank areas, and the arrangement of the elements in the graph is not very beautiful.

2. The dataset used in the article is not very large, and the test on the 20% dataset is not convincing. After all, it is a medical diagnosis, and you should look for more convincing data sets as much as possible.

3. Why finally trained 18 3D Vision Transformers on T2-weighted axial acquisitions? You should verify that several VIT combinations can show the best performance.

4. Some related newly works and surveys should be introduced and cited. For example:

[1]He, Kelei, et al. "Transformers in medical image analysis: A review." Intelligent Medicine (2022).

[2]Golfe, Alejandro, et al. "ProGleason-GAN: Conditional Progressive Growing GAN for prostatic cancer Gleason Grade patch synthesis." Computer Methods and Programs in Biomedicine (2023): 107695.

[3] Li, Zewen, et al. "A survey of convolutional neural networks: analysis, applications, and prospects." IEEE transactions on neural networks and learning systems (2021).

Reviewer 3 Report

It is interesting study with state-the-art technique applied. Here is my comments below.

1. Figure 1, patch is not shown in volume form. The font size and arrange should be improved and enhanced

2. Font and art object are two small Figure 2. Suggest to redesign Figure 1 and 2.

3. Sample Image/figures must be given for up-sampling, down-sampling, and cropped image along with the original image.

4. In section 3.4.2, example of CSP and CSE should be included more specific related to this study. The CSP and CSE need to more detail and clarify.

5. The formula for calculation two-/three-model should be shown.

6. In Table 6 and 8, the highest and lowest in the column should be highlighted.

Round 2

Reviewer 1 Report

I carefully read the revised version of this manuscript. As can be understood, my questions are clarified, and previous issues are resolved. This manuscript is suitable for acceptance.